# Structure and function of yeast Lso2 and human CCDC124 bound to hibernating ribosomes

Jennifer N. Wells[1ʘ], Robert Buschauer[1ʘ], Timur Mackens-Kiani[1], Katharina Best[1], Hanna Kratzat[1], Otto Berninghausen[1], Thomas Becker[1], Wendy Gilbert[2], Jingdong Cheng[1]*, Roland Beckmann[1]*

**1** Gene Center and Center for Integrated Protein Science Munich, Department of Biochemistry, University of Munich, Munich, Germany, **2** Department of Molecular Biophysics and Biochemistry, Yale University, New Haven, Connecticut, United States of America

ʘ These authors contributed equally to this work.
* jcheng@genzentrum.lmu.de (JC); beckmann@genzentrum.lmu.de (RB)

**Data Availability Statement:** All molecular models files are available from the PDB with the following PDB IDs: native Lso2-80S: 6Z6J reconstituted Lso2-80S: 6Z6K, CCDC124-80S: 6Z6L, eEF2/

## Abstract

Cells adjust to nutrient deprivation by reversible translational shutdown. This is accompanied by maintaining inactive ribosomes in a hibernation state, in which they are bound by proteins with inhibitory and protective functions. In eukaryotes, such a function was attributed to suppressor of target of Myb protein 1 (Stm1; SERPINE1 mRNA-binding protein 1 [SERBP1] in mammals), and recently, late-annotated short open reading frame 2 (Lso2; coiled-coil domain containing short open reading frame 124 [CCDC124] in mammals) was found to be involved in translational recovery after starvation from stationary phase. Here, we present cryo-electron microscopy (cryo-EM) structures of translationally inactive yeast and human ribosomes. We found Lso2/CCDC124 accumulating on idle ribosomes in the nonrotated state, in contrast to Stm1/SERBP1-bound ribosomes, which display a rotated state. Lso2/CCDC124 bridges the decoding sites of the small with the GTPase activating center (GAC) of the large subunit. This position allows accommodation of the duplication of multilocus region 34 protein (Dom34)-dependent ribosome recycling system, which splits Lso2-containing, but not Stm1-containing, ribosomes. We propose a model in which Lso2 facilitates rapid translation reactivation by stabilizing the recycling-competent state of inactive ribosomes.

## Introduction

Ribosomes are universally conserved biological machines that translate genetic information from mRNA templates into polypeptides with the corresponding amino acid sequence. Maintaining ribosome functionality is vital for cell survival under all circumstances. Hence, regulatory mechanisms have evolved that facilitate transition of the translation machinery between active and dormant states. This allows cells to dynamically adapt to environmental changes in nutrient availability.

SERBP1-80S: 6Z6M, EBP1-80S: 6Z6N). All cryo-EM maps are available from the EMD (native Lso2-80S: EMD-11096, reconstituted Lso2-80S: EMD-11097, CCDC124-80S: EMD-11098, eEF2/SERBP1-80S: EMD-11099, EBP1-80S: EMD-11100).

**Funding:** Funding was provided by the German Research Council grants GRK1721 and FOR1805 (https://www.dfg.de/) for R. Beckmann. We acknowledge support by the Center for Integrated Protein Science Munich (CiPS-M). R. Buschauer is supported by a Boehringer Ingelheim Fonds PhD Fellowship, and JW is part of the International Max Planck Research School Life Science. The funders had no role in study design, data collection and analysis, decision to publish, or preparation of the manuscript.

**Competing interests:** The authors have declared that no competing interests exist.

**Abbreviations:** A, acceptor; ABC, ATP binding cassette; ABCE1, ATP binding cassette subfamily E member 1; Arx1, associated with ribosomal export complex protein 1; CCDC124, coiled-coil domain containing short open reading frame 124; cryo-EM, cryo-electron microscopy; CTF, contrast transfer function; CV, column volume; Dom34, duplication of multilocus region; D.m., *Drosophila melanogaster*; E, exit; EBP1, ErbB3-binding protein 1; eEF2, eukaryotic elongation factor 2; eIF6, eukaryotic initiation factor 6; ePAR-CLIP, Photoactivatable-Ribonucleoside-Enhanced Crosslinking and Immunoprecipitation with enhanced method of CLIP library preparation; GAC, GTPase activating center; Hbs1, Hsp70 subfamily B suppressor; HEK, human embryonic kidney; HPF, hibernation promoting factor; H.s., *Homo sapiens*; IFRD2, interferon-related developmental regulator 2; LHPF, long form hibernation promoting factor; Lso2, late-annotated short open reading frame 2; MetAP, methionine aminopeptidase; NatA, Nα-Acetyltransferase A; P, peptidyl; pY, protein Y; RAC, ribosome associated complex; RaiA, ribosome associated inhibitor 1; RBF, ribosomal binding factor; Rli1, RNase L inhibitor 1; RMF, ribosome modulation factor; Sec61, secretory protein 61; SERBP1, SERPINE1 mRNA-binding protein 1; SRP, signal recognition particle; Stm1, suppressor of target of Myb protein 1; S.c., *Saccharomyces cerevisiae*; β-ME, β-mercaptoethanol.

Nutrient-starvation–induced cellular stress and reversible translational repression is particularly well studied in prokaryotes. In bacteria, several small ribosomal binding factors (RBFs) have been identified that inhibit translation and facilitate reversible formation of inactive 100S ribosome dimers [1,2]. The formation of 100S dimers allows ribosomes to enter a hibernation state during stationary growth or stress phases [3–5]. Within hibernating bacterial ribosomes, RBFs (ribosome modulation factor [RMF], hibernation promoting factor [HPF], and ribosome associated inhibitor A [RaiA]) bind the decoding center, occupy the mRNA binding channel, and block acceptor (A) and peptidyl (P) tRNA sites [6–8]. As a result, bacterial hibernation factors protect the crucial active sites of the ribosome and inhibit binding of both mRNA and tRNA, blocking translation altogether [9].

Several types of inactive or hibernating ribosomes have also been observed in eukaryotes [10,11]. In general, idle 80S ribosomes accumulate in eukaryotic cells after exposure to various stresses like amino acid shortage [10,12], osmotic stress [13], or glucose starvation [14,15]. To restart translation after stress, these 80S need to be dissociated in order to repopulate the pool of free 40S for initiation. The duplication of multilocus region 34 (Dom34) splitting system, containing the termination factor homologs Dom34 (Pelota in mammals), Hsp70 subfamily B suppressor 1 (Hbs1), and the ATP binding cassette (ABC)-type ATPase RNase L inhibitor 1 (Rli1; ATP binding cassette subfamily E member [ABCE1] in mammals), were shown to dissociate vacant ribosomes in yeast [15] and mammals [16]. The first eukaryotic hibernation factor, suppressor of target of Myb protein 1 (Stm1), was found in the crystal structures of otherwise empty yeast 80S ribosomes, which were prepared from cells following 10 min of glucose starvation [17]. Stm1 and its mammalian homolog SERPINE1 mRNA-binding protein 1 (SERBP1) clamp the ribosomal subunits together and, similar to bacterial hibernation factors, bind in the mRNA entry channel as well as the A and P sites of the 40S, thereby occupying sites important for translational activity[17].

The nonessential protein Stm1 was shown to have a protective role in *Saccharomyces cerevisiae* (*S.c.*), in which it supports recovery of translation after quiescence [18,19], and a knockout of Stm1 in yeast suppresses the requirement of the Dom34 splitting system to restart translation after glucose starvation [15]. This indicates that idle 80S lacking Stm1 are less stable in vivo. Metazoan homologs of Stm1 were also observed bound to inactive ribosomes from *Drosophila melanogaster* (*D.m.*) and *Homo sapiens* (*H.s.*), indicating a high degree of functional conservation [20]. Notably, these ribosomes also contain tRNA in the ribosomal exit (E) site and the eukaryotic elongation factor 2 (eEF2) [20]. Presence of eEF2 and E-site tRNA was later also observed on inactive ribosomes derived from rabbit reticulocyte lysates [11]. In the same system, another type of inactive ribosomes was found containing the poorly characterized protein interferon-related developmental regulator 2 (IFRD2) and tRNA in a newly defined position, near the E site, termed the Z site [11]. However, formation, release, and molecular function of the involved RBFs remain enigmatic for all types of hibernating eukaryotic ribosomes.

Recently, another eukaryotic RBF responsible for protecting and recovering translation was discovered: late-annotated short open reading frame 2 (Lso2) in *S.c.*, which is highly homologous to coiled-coil domain containing short open reading frame 124 (CCDC124) in human cells [21]. Based on chemical crosslinking studies, Lso2 constitutively associates with 80S monosomes and binds in proximity to tRNA and to rRNA helices H43 and H44 [21]. This interaction between Lso2 and ribosomes apparently facilitates reactivation of translation upon nutrient upshift and exit from stationary phase. Lack of Lso2 appears to affect translation at the stage of initiation, as evident from a global 4- to 5-fold reduction in ribosome association with most mRNAs in *LSO2* knockout cells (*lso2Δ*) during recovery. Ribosomes from *lso2Δ* show additional functional defects, including altered sensitivity to RNase I and altered A-site

tRNA accommodation, as determined by increased incidence of pausing at start codons and enrichment of 21mers in ribosomal profiling, indicative of empty A sites [21,22]. The mode of ribosome interaction, and thus the molecular basis for these effects of Lso2, is not understood.

Here, we have structurally characterized the interaction between Lso2 and CCDC124 with eukaryotic 80S ribosomes by single-particle cryo-electron microscopy (cryo-EM) to illuminate how binding of these proteins can modulate translational activity during and after starvation. We reconstituted Lso2 with idle 80S ribosomes from purified components and also characterized native idle 80S ribosomes obtained from yeast grown under nutrient-limiting conditions in minimal medium and from human (human embryonic kidney [HEK]293T) cells harvested at high confluency. High-resolution structures of the yeast Lso2–80S and the human CCDC124–80S complexes reveal near-identical binding modes of Lso2 and CCDC124 to ribosomes. Within idle ribosomes, these factors occupy the P-site position of the 40S subunit, including mRNA and tRNA binding sites. Moreover, they bind the 60S subunit in the A and P sites and reach close to the stalk base and the GTPase activating center (GAC) [23]. Surprisingly, we find that a majority of human 80S are occupied with ErbB3-binding protein 1(EBP1)—a homolog of the ribosome biogenesis factor "associated with ribosomal export complex protein 1" (Arx1)—bound to the peptide tunnel exit of the 60S [24]. Notably, we observe, in addition to the class containing nonrotated CCDC124-bound 80S, the previously described idle 80S bound to SERBP1 and eEF2 in the rotated state. These human SERBP1/eEF2 80S are similar to the inactive Stm1-bound ones in the crystal structures of yeast ribosomes [17], suggesting that in eukaryotes, at least 2 functionally different pools of idle ribosome exist. Exploring their functional distinction, we show that only Lso2-containing, but not Stm1-containing, idle 80S can be readily split by the Dom34 splitting system. This strongly suggests a function of Lso2 in providing a pool of easily recyclable 80S for quick resumption of translation after nutrient starvation.

## Results

### Identification of Lso2 bound to idle yeast 80S ribosomes

For structural analysis of Lso2-bound ribosomes, we reconstituted idle 80S ribosomes from *S. c.* in vitro with a 10× molar excess of purified recombinant Lso2 under defined conditions (see Materials and Methods) and performed single-particle cryo-EM. From 3D classification (S1A Fig), we obtained 1 class displaying a nonribosomal helix-shaped extra density between the large and small subunit (Fig 1A). After refinement, we obtained a structure at 3.4 Å average resolution and local resolution for Lso2 ranging from 3.2–4.5 Å (Fig 1A, left, and S1D Fig), which allowed us to build an atomic model for Lso2 (Fig 1A, middle, S2 Fig and S1 Table).

Lso2 is exclusively bound to the inactive 80S in the unratcheted/nonrotated conformation, as, for example, observed in 80S ribosomes stalled with an empty A site [25] (Fig 1B). This is unusual because idle yeast 80S have normally been observed primarily in rotated states (Fig 1C), independent of the presence of Stm1 [17,28,29]. In our structure, Lso2 comprises 2 α-helices connected by a short loop (S1 and S2 Figs). The first Lso2 helix is located in the intersubunit space between small and large subunit. More specifically, it occupies the P-tRNA binding site and the mRNA channel in the P and E sites of the small subunit. (Fig 1D). From there, Lso2 bridges over towards the A site of the 60S subunit, whereby the second helix stretches below the central protuberance and extends towards the GAC and the stalk base. In this position, Lso2 would overlap with an accommodated A-site tRNA (Fig 1D). In conclusion, Lso2 occupies mRNA and tRNA binding sites on the ribosome that need to be accessible for translation, thus corroborating Lso2's function as a ribosome hibernation factor.

Confirming the reconstituted Lso2–80S complex structure, we determined the native structure of Lso2-bound 80S ribosomes (Fig 1A, right), which we found in yeast cells cultured in

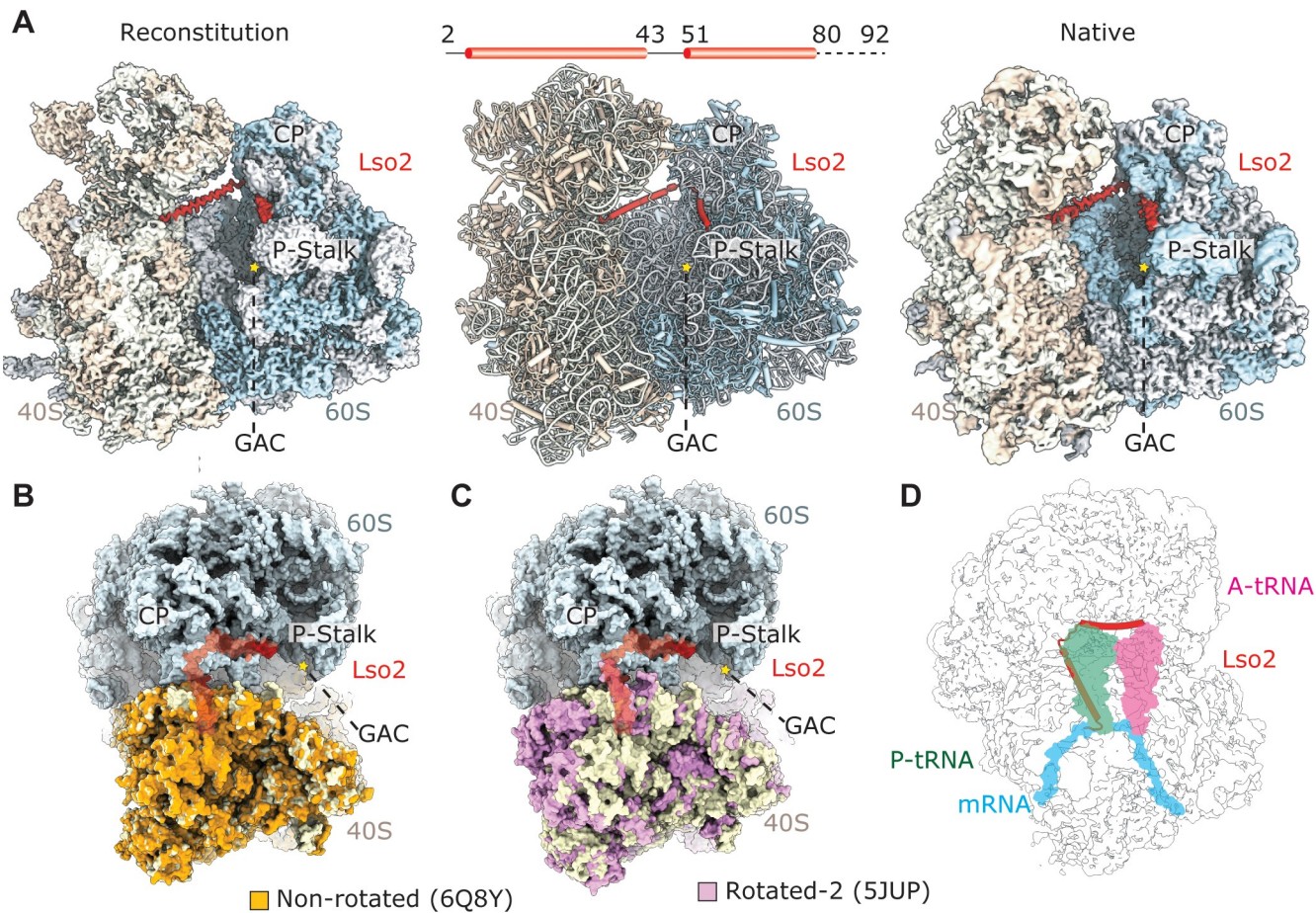

**Fig 1. Identification of Lso2 bound to idle 80S ribosomes.** (A) Left: 3.4 Å resolution cryo-EM map of the in vitro reconstituted yeast Lso2–80S complex; α-helical extra density for Lso2 (red) is found in the intersubunit space. Middle: atomic model of the Lso2–80S complex. Right: 3.5 Å resolution cryo-EM map obtained from a native 80S ribosome preparation from cells grown under nutrient-limiting condition. (B–D): Top views of the Lso2–80S structure superimposed with the structure of a yeast 80S in the nonrotated state (B) (PDB: 6Q8Y [25]) and a yeast 80S in the rotated-2 state (C) (PDB: 5JUP [26]), showing differences between small subunit 18S rRNA in the 2 rotational states. (D) Same view as C, illustrating the position of accommodated A- and P-site tRNAs and mRNA (PDB: 5MC6) [27]. All structures were aligned on the 60S subunit. Hallmark features of the 80S ribosome are labelled. A, acceptor; CP, central protuberance; cryo-EM, cryo-electron microscopy; GAC, GTPase activating center; Lso2, late-annotated short open reading frame 2; P, peptidyl; PDB, Protein Data Bank.

minimal medium. Extensive classification (S1B Fig) yielded a structure of an Lso2–80S complex at 3.5 Å resolution that was essentially indistinguishable from the structure from the in vitro reconstituted complex (S2 Fig and S1 Table). We conclude that also in vivo, Lso2 preferably binds to empty nonrotated 80S ribosomes in the same conformation as observed upon in vitro reconstitution.

## Lso2 interacts with ribosomal tRNA and mRNA binding sites

The local resolution in the intersubunit space allowed us to unequivocally identify Lso2 and describe its ribosome interactions based on side-chain resolution (Figs 2 and S2). The main interaction of Lso2 with the 40S subunit is formed by the positively charged Lso2 N-terminus that reaches into the mRNA channel of the P and E sites (Fig 2A). Here, Lso2 residues G2 to S6 interact in the cleft between rRNA helices h24, h28, and h44 (Fig 2A). In detail, Lso2 F5 is stacking with the bases G1150 of h28 and G1768 of h44, an interaction likely supported by

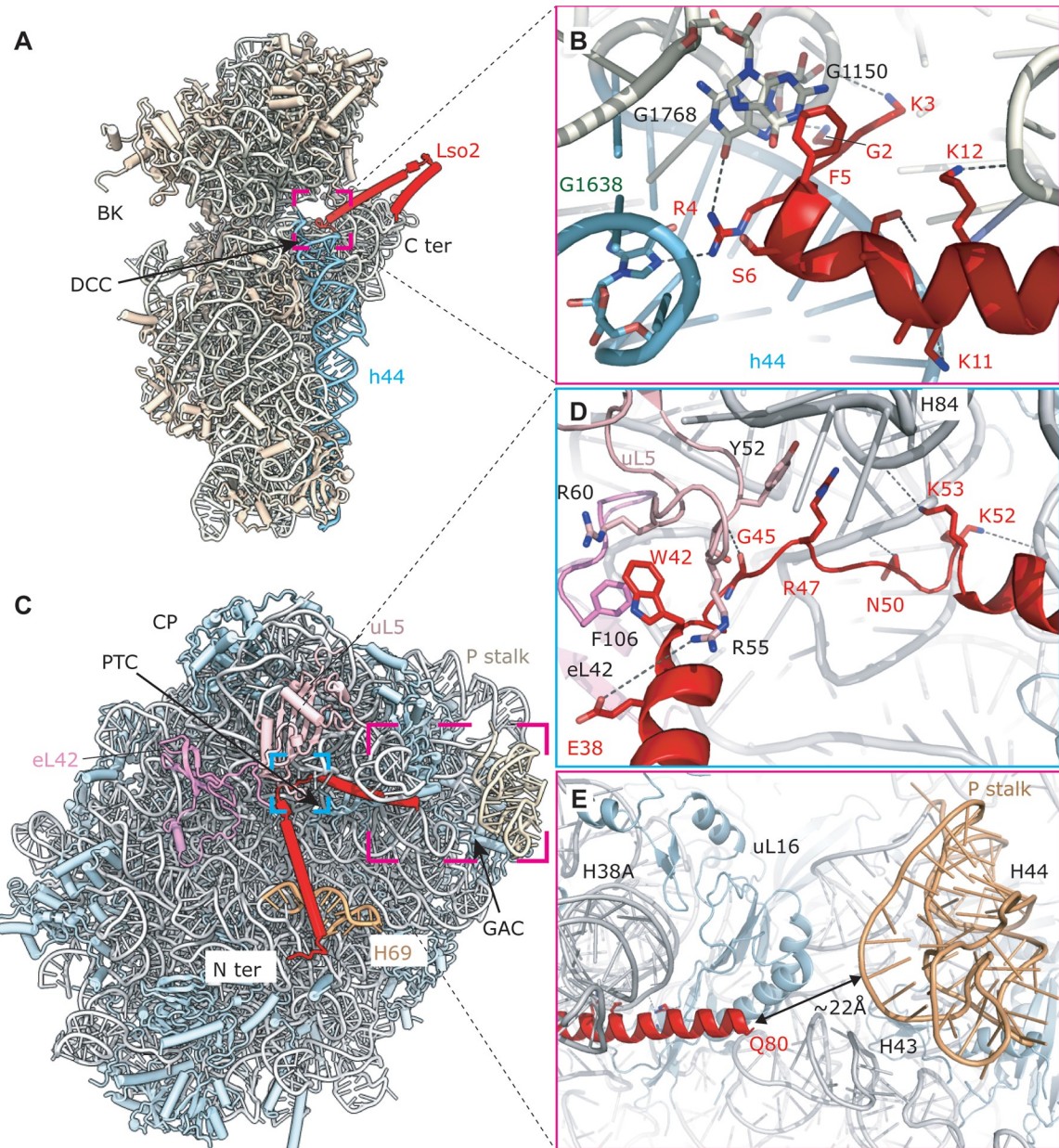

**Fig 2. Lso2 interactions with the ribosome.** (A) Positioning of Lso2 with respect to the small subunit. (B) Zoom on interactions of Lso2 in the P and E sites of the 40S: the Lso2 N-terminus reaches into the cleft between rRNA helices h24, h28, and h44 of the 18S rRNA. All the hydrogen-bond and salt-bridge interactions are indicated with a dashed line. (C) Positioning of Lso2 with respect to the large subunit. (D) Lso2 binds to the large subunit near the CP via the helix-connecting loop. Interactions involve ribosomal proteins uL5 and eL42 as well as H84 and H85 of 25S rRNA. (E) The C-terminal α-helix of Lso2 passes by the major groove of H38A and uL16 towards the stalk base formed by H43 and H44 (GAC) and the P-stalk. The ultimate C-terminus of Lso2 is missing but might be located very close to the P-stalk. BK, beak; CP, central protuberance; DCC, Decoding center; E, exit; GAC, GTPase activating center; Lso2, late-annotated short open reading frame 2; P, peptidyl; PTC, peptidyl transferase center.

several salt bridges and hydrogen bonds formed between highly enriched, positively charged Lso2 residues (K3, R4, K11, K12) and the negatively charged 18S rRNA (Fig 2B). The Lso2 N-terminus thus occupies the site where in an active ribosome, the last 2 mRNA bases from the E-site codon and the first mRNA base of the P-site codon would be located.

Protruding from the 40S P site, the density for the first Lso2 helix ends near the central protuberance of the large subunit (Fig 2C). The connecting loop constitutes an important interface involving ribosomal proteins uL5 and eL42, as well as 25S rRNA. In detail, Lso2 W42 is accommodated in a pocket formed by R55 and R60 from uL5 and F106 from the C-terminus of eL42 (Fig 2D). Further interactions between Lso2 residues E38, G45, and R47 with uL5 Y52 and R55, as well as between Lso2 N50, K52, and K53 and the phosphate backbone of 25S rRNA helices H84 and H85, are well resolved (S2C and S2D Fig). The second straight helix of Lso2 continues along the major groove of rRNA helix H38A (also known as the A-site finger) and uL16 towards the stalk base and the P-stalk (Fig 2E). Notably, both the A-site finger and uL16 are contact sites for the elbow region of A-site tRNA during accommodation and translocation [30–32]. In our structure, these contact sites are blocked by Lso2. Photoactivatable-Ribonucleoside-Enhanced Crosslinking and Immunoprecipitation with enhanced method of CLIP library preparation (ePAR-CLIP) suggested direct interaction of Lso2 with rRNA helices H43/H44 of the GAC [21]. This interaction is most likely established by the ultimate C-terminus of Lso2, which is not resolved in our maps because of a high degree of flexibility.

## Two distinct, inactive ribosomal species are present in eukaryotes

Because Lso2 is conserved in higher eukaryotes [21], we expanded our studies on its role as a starvation factor to the human system. To that end, human 80S ribosomes were prepared from an HEK293T culture with high cell density. Mass spectrometry confirmed the presence of Lso2 homolog CCDC124 as well as SERBP1, eEF2, and EBP1. CCDC124 and EBP1 were also detected in western blot analysis (S3 Fig).

Cryo-EM analysis of this sample revealed 2 major classes of hibernating 80S ribosomes (S4A Fig). One class showed helical density very similar to the density for Lso2 on the yeast ribosome, which again adopted the nonrotated state. As expected, the density corresponded to CCDC124. Interestingly, the same class also displayed density corresponding to EBP1 at the peptide exit site [33–35], as well as tRNA in the E site. The second class also contained EBP1 and E-site tRNA, but instead of CCDC124, SERBP1 and eEF2 were present, and ribosomes were stabilized in the rotated state, as previously observed in cryo-EM structures of inactive 80S ribosomes from *D.m.* and *H.s.* [11,20].

Both classes, as well as a merged class of all EBP1-containing particles, were refined independently (S4B–S4D Fig), and a molecular model was built for the human CCDC124–EBP1–80S, an SERBP1–eEF2–EBP1–80S complex, and an EBP1–80S complex for which all EBP1-containing particles were refined (Figs 3A and 3B and S5 and S1 Table).

## The ribosome binding mode of Lso2 and CCDC124 is highly conserved

As for the yeast Lso2–80S complex, the local resolution of the human structure was sufficient to assign the helical density unambiguously to CCDC124 and describe its ribosome interactions (Figs 3E and S4B–4D and S5 and S1 Table). Position and orientation of CCDC124 in the A and P sites of the 40S was near identical compared to the yeast homolog, indicating a conserved binding mode (Fig 3). Ribosome binding is mediated by residues conserved from *S.c.* to *H.s.* (Fig 3C), including W39 (W42 in *S.c.*), which plays an important role in establishing the main contact to the 60S in the P site.

In detail, the N-terminal helix of CCDC124 occupies a similar space as Lso2 in the P- and E-site mRNA position (Fig 3D), yet the electron density for the N-terminal helix was weaker than for the C-terminal one, most likely because of higher overall flexibility of the 40S subunit. The resolution of 3.0 Å was sufficient for molecular model building starting from K11. As with Lso2, the first helix stretches over towards the 60S subunit. At the connecting loop, W39 of CCDC124

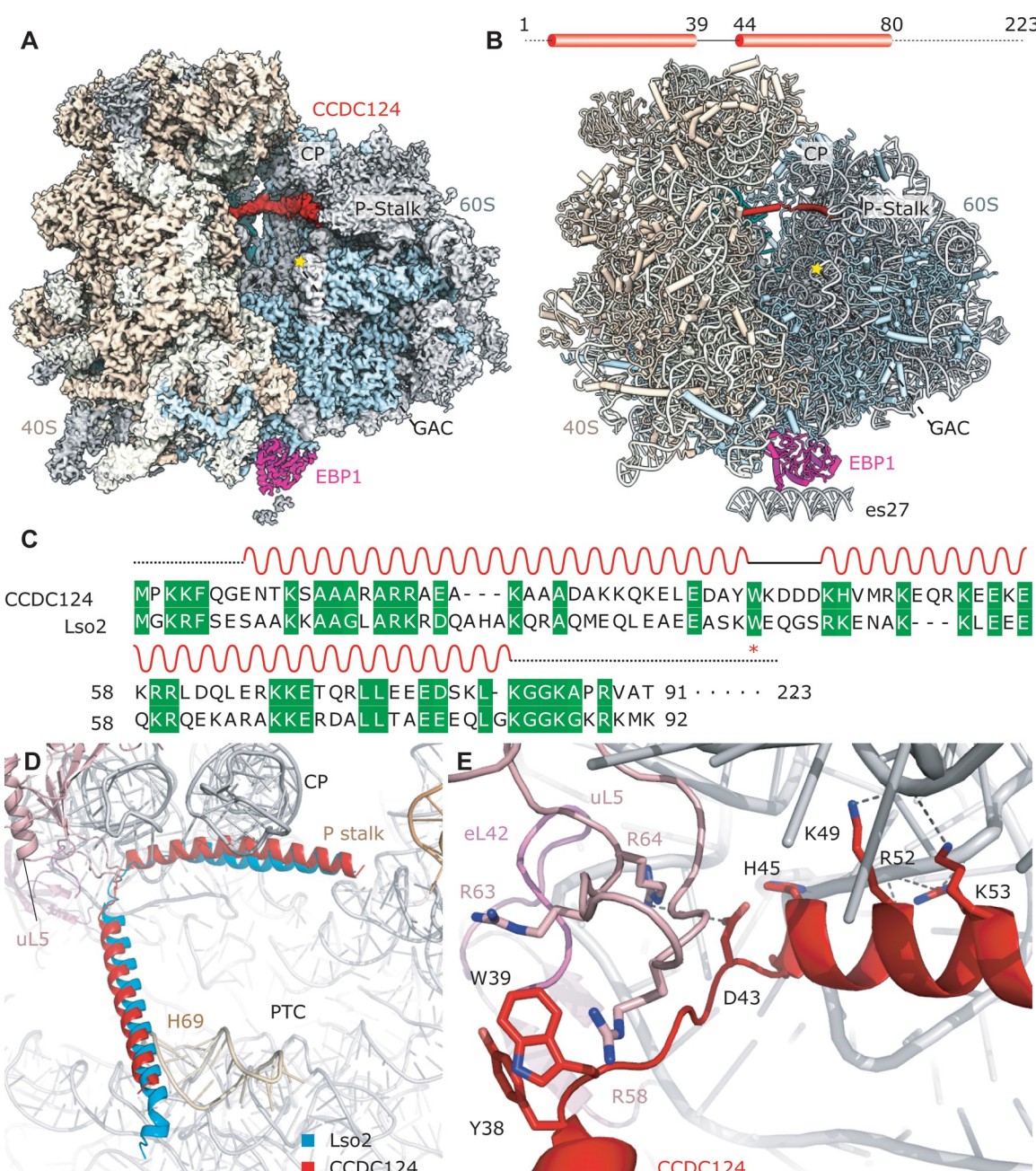

**Fig 3. Ribosome binding of Lso2 and CCDC124 is evolutionarily conserved.** (A) 3.0 Å resolution cryo-EM map of the human CCDC124–EBP1–80S ribosome. The reconstruction also contains E-site tRNA. The CCDC124 density is displayed at a different contour level than the ribosome map. (B) Atomic model of the CCDC124–EBP1–80S complex. (C) Conserved residues shown in a sequence alignment of CCDC124 with Lso2. (D) Superposition of Lso2–80S with CCDC124–80S models based on 25S rRNA shows a very similar positioning, indicating a highly conserved binding mode. CCDC124, coiled-coil domain containing short open reading frame 124; CP, central protuberance; cryo-EM, cryo-electron microscopy E, exit; EBP1, ErbB3-binding protein 1; GAC, GTPase activating center; Lso2, late-annotated short open reading frame 2; P, peptidyl; PTC, peptidyl transferase center.

(similar to W42 in Lso2) is accommodated on the 60S in a binding pocket formed by uL5 R58 and R63 (S5C Fig). Unlike the yeast Lso2 contacts, however, eL42 is not involved in CCDC124 binding (Fig 3E). The stacking interaction between W42 of Lso2 and F106 of eL42 is accordingly

not conserved in the human complex. Yet, it is compensated for by stacking between Y38 and W39 of CCDC124. In general, the helix-connecting loop adopts a slightly different path, though the second helix is observed in a nearly identical position as in Lso2 (Fig 3D). Analogous to what we observe in the Lso2 structures, presence of CCDC124 prohibits A-site tRNA accommodation, and again, the extended C-terminal tail is not resolved (Figs 3D and S5D). In conclusion, CCDC124 occupies the binding interfaces of A- and P-site tRNA, as well as mRNA on the human ribosome, congruent with the Lso2 binding site in yeast (Figs 3D and 1D).

The second major class of hibernating human ribosomes we observed and refined to 3.1 Å was in agreement with previous reports on structures in the rotated state, more specifically the "rotated-2 state," as occurring when the 80S is occupied with hybrid A/P and P/E tRNAs (Fig 1C) [26]. In addition to a type-II tRNA displaying an extended V-loop in the E site, it contains eEF2 and SERBP1 (Fig 4). Like Stm1, SERBP1, together with eEF2, binds in the mRNA entry channel and prevents mRNA binding in the A and P sites [11,17–20]. Importantly, binding of SERBP1 and CCDC124 is mutually exclusive.

## EBP1 binds to the peptide exit site of hibernating human ribosomes

We note that under our conditions, the human ribosomes happen to be stably associated with EBP1 at the tunnel exit (Figs 4 and S6), independent of ribosomal state and factors occupying the mRNA and tRNA binding sites. EBP1 is the human homolog of the yeast ribosome biogenesis factor Arx1 [34,36] which also binds to the ribosomal exit site and is related to methionine aminopeptidases (MetAPs) [36,37]. In addition, EBP1 has been described to be involved in cell proliferation [38] and human cancer [39].

We obtained a structure of EBP1–80S at 2.9 Å overall resolution, showing that EBP1 is anchored to the peptide exit site through ribosomal proteins uL23, eL19, and uL29, as well as 28S rRNA helix H59 (Figs 4C and S6). Also similar to Arx1 in yeast, it coordinates a part of the flexible rRNA expansion segment ES27L in a defined position below the tunnel exit. Its previously described insertion domain (residues 250–305) [36] interacts with rRNA helix H59 in a rearranged conformation and reaches into a cleft between uL23 and H59 (S6A Fig). On the tip of rRNA helix H59, the base U2708 is flipped out and stacks with EBP1 F266 (Fig 4C), while C2709 is sandwiched between R263 of EBP1 and Q40 of eL19. These interactions are supported by

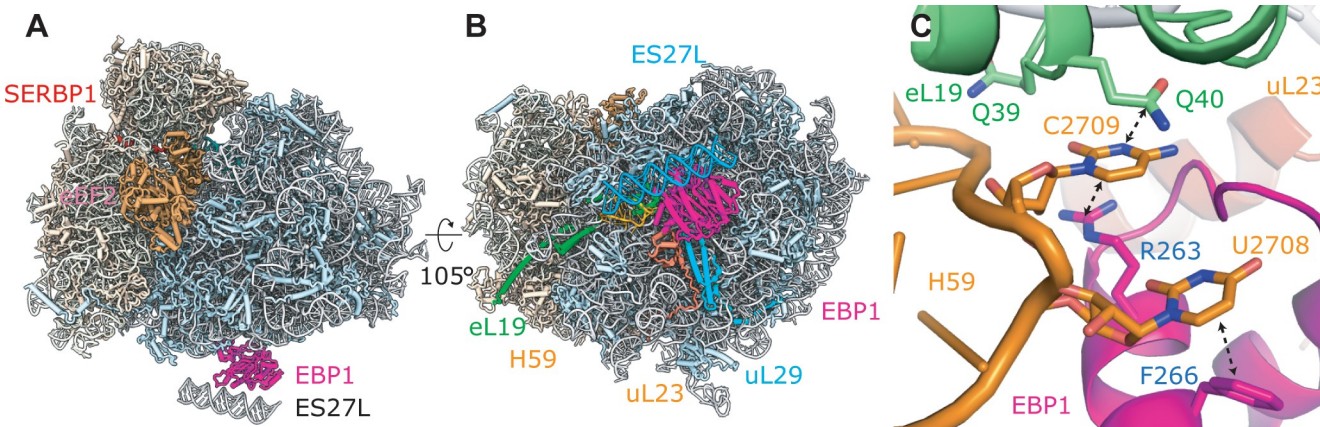

**Fig 4. Hibernating human ribosomes are bound to EBP1 on the peptide exit site.** (A) Side and bottom views of the human eEF2–SERBP1–EBP1–80S model, built from a 2.9 Å resolution cryo-EM map. (B) EBP1 coordinates a part of rRNA expansion segment ES27L below the peptide tunnel exit. A dummy RNA helix was used as a placeholder for ES27L. (C) Zoom on the EBP1 insertion domain reaching into a cleft between uL23 and H59. cryo-EM, cryo-electron microscopy; EBP1, ErbB3-binding protein 1; eEF2, eukaryotic elongation factor 2; SERBP1, SERPINE1 mRNA-binding protein 1.

additional contacts between its MetAP-like domain and uL23 and uL29 (Fig 4B). Interestingly, EBP1 shares a similar interaction mode with the ribosome as nascent chain-interacting factors such as ribosome associated complex (RAC), the signal recognition particle (SRP), secretory protein 61 (Sec61), or the N-α-Acetyltransferase A (NatA) complex [40–43], and the presence of EBP1 would not allow simultaneous binding of any of the abovementioned factors (S6C Fig).

## Lso2-bound, but not Stm1-bound, 80S are split by canonical recycling factors

Our cryo-EM analysis of human hibernating 80S suggests that in eukaryotes, at least 2 clearly distinguishable populations of idle, translationally repressed 80S exist. These 80S are either bound to Stm1/SERBP1 together with eEF2 or to Lso2 (CCDC124). Besides the different factor compositions, their main difference is the conformation of the ribosome: Stm1/SERBP1-containing 80S so far have been exclusively observed in rotated states [11,17,20,26], whereas the Lso2- or CCDC124-bound 80S were found in the nonrotated state (Fig 1B and 1C), similar to the post-translocational state with tRNAs in P and E sites and an empty A site [25,44]. Thereby, Lso2 (CCDC124) on 80S occupies a position that would exclude simultaneous presence of the basic translation machinery (A-site tRNA, P-site tRNA, mRNA) (Fig 1D) and concurrently stabilize the nonrotated ribosome conformation [26]. Notably, Dom34–Hbs1 and Dom34–ABCE1 complexes preferably bind to ribosomes in the nonrotated post state (Fig 5A and 5B) [45–49]. In the rotated state, which is stabilized by Stm1, these factors would clash with the ribosome itself (S7 Fig) and with eEF2, if present. We thus hypothesized that Lso2-bound 80S ribosomes are a substrate for the Dom34 splitting system and tested our hypothesis using an in vitro ribosome splitting assay. To that end, reconstituted Lso2–80S complexes were incubated with purified Dom34, Hbs1–GTP, ABCE1–ATP, and eukaryotoc initiation factor 6 (eIF6), a factor that prevents subunit reassociation [50,51]. Splitting reactions were subjected to sucrose density gradient centrifugation, and UV profiles of the gradients were generated at 260 nm (S8 Fig and S1 Data).

We observed that under near-physiological buffer conditions, reconstituted Lso2–80S ribosomes were almost quantitatively split, even more efficiently than puromycin/high-salt–treated empty 80S ribosomes (Figs 5D–5E and S9 and S2 Data).

Splitting of Lso2–80S was dependent on the presence of nucleotides (1 mM ATP, 1 mM GTP), as well as Dom34 and ABCE1 [15,52]. Also consistent with these studies is the observation that Hbs1, although required in vivo [15], was not required for ribosome splitting in vitro [16]. This indicated that indeed the enzymatic activity of ABCE1 in concert with Dom34 binding to the A site was required to split Lso2-bound 80S ribosomes (S10 Fig and S3 Data). Although the presence of eEF2 would not allow for the Dom34–ABCE1 system to directly act on Stm1/SERBP1-containing 80S, we wondered whether splitting by the Dom34 system depends on the nonrotated conformation induced by Lso2. To that end, we tested Stm1-containing yeast 80S ribosomes obtained after glucose starvation, which are known to be in the rotated state and have a tendency to lose the majority of the bound eEF2 during centrifugation [17]. As anticipated, Stm1–80S complexes were found to be essentially resistant to splitting by the Dom34 system, arguing in favor of a nonrotated conformation being required for efficient splitting by Dom34–ABCE1 (Figs 5D–5E and S9 and S1 and S2 Datas). Notably, Stm1–80S complexes purified after glucose starvation may not be directly comparable to in vitro reconstituted Lso2–80S complexes. Therefore, we cannot entirely exclude other differences than the rotation status of Stm1–80S contributing to the splitting resistance.

Thus, to check whether the splitting-resistant fraction in the Stm1–80S population contained Stm1 bridging the 40S and 60S in a rotated conformation, we performed cryo-EM on

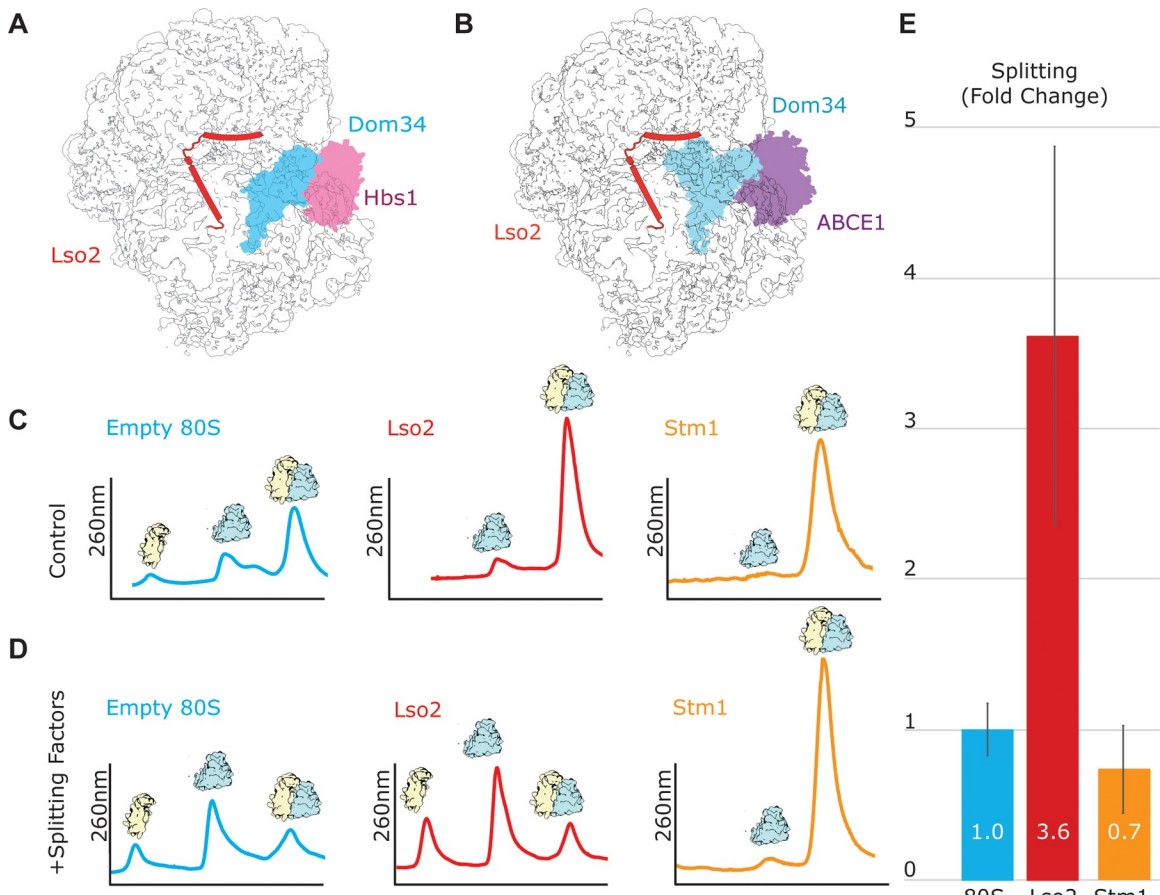

**Fig 5. In vitro splitting assay with Lso2–80S and Stm1–80S.** (A and B) The Lso2–80S in nonrotated state (displayed as cartoon) was overlaid with ribosome rescue factors Dom34 and Hbs1 (PDB: 3IZQ) [45] or Dom34 and ABCE1 (PDB: 3J16) [46]. In both conformations, Dom34 could accommodate within the A site of the 40S and would not clash with Lso2, leading to the hypothesis that the Dom34 splitting system preferably splits Lso2–80S. (C and D) Representative sucrose gradient profiles from splitting assays carried out without (C) or with (D) splitting factors; raw data can be found in S1 Data. (E) Quantification of relative splitting displayed as fold change over the negative controls (experiments without splitting factors), normalized to the control experiment using puromycin-treated empty 80S (see also S8 and S9 Figs and S2 Data). All experiments were performed in triplicates. A, acceptor; Dom34, duplication of multilocus region 34; Hbs1, Hsp70 subfamily B suppressor; Lso2, late-annotated short open reading frame 2; PDB, Protein Data Bank; Stm1, suppressor of target of Myb protein 1.

the Stm1–80S fraction after the splitting reaction with and without splitting factors. In any case, we observed that the vast majority (>90%) of splitting-resistant 80S adopted the same "rotated-2 state" [26] as the human eEF2 and SERBP1 containing ribosomes, and in the majority of these ribosomes, Stm1 could be clearly observed in the subunit-bridging conformation crossing the mRNA channel (S11 Fig). In conclusion, these results strongly suggest that, in contrast to the rotated Stm1/SERBP1 (and eEF2)-containing 80S, the nonrotated Lso2/CCDC124-bound ribosomes are substrates, which can be efficiently recycled by the Dom34–splitting system.

## Discussion

While for bacteria, numerous hibernation factors have been characterized (for review, see, for example, [9]), knowledge about eukaryotic equivalents was so far limited to Stm1 in yeast and SERBP1/eEF2, as well as the recently discovered IFRD2/Z-site tRNA [11,17] in mammals. In

analogy to bacterial counterparts, both SERBP1/eEF2 and IFRD2/Z-tRNA occupy crucial active sites on the ribosome [11]. While structural data strongly suggested that hibernation factors act by maintaining ribosomes in a translation-incompetent state, the mechanisms of recovering translation-competent ribosomes have remained enigmatic.

In this work, we present Lso2 and CCDC124 in yeast and human cells as new eukaryotic-specific ribosome hibernation factors that play an active role in translation recovery [21]. Our cryo-EM structures show that Lso2 and CCDC124 occupy the binding sites of mRNA and tRNA, thereby resembling the mode of action of known bacterial hibernation factors. However, while RMF, HPF, RaiA (also known as protein Y or pY), or the long form hibernation promoting factor (LHPF) N-terminal domain all target the mRNA path and decoding center of the small subunit, Lso2 and CCDC124 bind to both 40S and 60S subunits and exclusively stabilize the nonrotated conformation of the 80S ribosome.

Independent of the ribosomal state and binding of hibernation factors SERBP1 or CCDC124, we observed EBP1 bound to the peptide tunnel exit of hibernating human ribosomes. Our EBP1–80S structure largely resembles the recently published structures of EBP1 bound to puromycin-treated 80S ribosomes [53] and of EBP1-bound ribosomes from the mouse neocortex [54]. The binding mode we observed is similar to that of other exit site binding proteins that interact with the nascent chain. As a result, this binding mode excludes interaction of the majority of co-translationally acting factors to the ribosome, consistent with a role as hibernation factor that prevents unproductive sequestration of these factors to idle ribosomes. Interestingly, the study by Kraushar and colleagues showed that EBP1 binds not only idle 80S but also to translating polysomes [54]. Here, EBP1 was found highly enriched in the cytosol of early-born neuronal stem cells of mouse neocortex, in which it has been suggested to play a role in modulating protein homeostasis, possibly by competing with other exit-site factors. In this context, it is intriguing that EBP stabilizes the otherwise flexible rRNA expansion segment ES27 in an exit-site–facing conformation. While no data are available for the human system, ES27 is essential in yeast and was shown to directly coordinate and position nascent chain-interacting factors such as NatA [43] on the exit site. Another potential role of EBP1 may be in preventing ubiquitination of the ribosomal protein uL29 of the inactive ribosome, thereby contributing to the coordination of its propensity for ribophagy [55]. Yet, the exact circumstances and mechanism of how EBP1 is recruited to dormant and also translating ribosomes and what triggers its dissociation before or during the translation cycle remain to be elucidated.

We reasoned that keeping dormant ribosomes in the nonrotated state may be a prerequisite for the preferred reactivation by the Dom34 splitting system. Indeed, using in vitro splitting assays, we could show that Lso2-bound 80S are highly susceptible to recycling. In contrast, 80S enriched in Stm1, as occurring after short glucose starvation [17], could not be split, which is explained by our structures of Stm1–80S resembling a distinct rotated-2 state [26]. Moreover, presence of SERBP1 on the ribosome results to a large extent in stable eEF2 association in mammals [11,20]. An analogous eEF2 enrichment on Stm1-bound ribosomes has been shown, at least in vitro, in yeast [56]. This would further block access of Dom34 to the ribosomal A site and render these ribosomes resistant to splitting by this system.

These findings are also in good agreement with the initial observation that absence of Lso2 delays recovery of translation after starvation [21]. Moreover, Lso2-depleted cells accumulate idle 80S ribosomes and are decreased 5-fold in global initiation as analyzed by ribosome profiling [21]. Intriguingly, a study demonstrated that deletion of Dom34 and Hbs1 has similar effects on recovery from glucose depletion in yeast as deletion of Lso2 and that idle ribosomes isolated from glucose-starved yeast were substrates for splitting by the Dom34 system [15]. Along the same lines, deletion of the stabilizing Stm1 suppresses the requirement of Dom34

and Hbs1 in recovery of translation [15]. Here, a likely explanation is that in the absence of Stm1, there is a larger pool of idle ribosomes that are unstable and can be easily recycled. Thus, recovery of translation depends to a lesser extent on the recycling of ribosomes that strictly require the Dom34 system for splitting. In support of this antagonism between Stm1 and Dom34, overexpression of Stm1 in a *dom34Δ* strain leads to a growth defect [19], likely due to an accumulation of Stm1–80S and the inability to split the remaining pool of 80S by the Dom34 system. In combination with these data, our findings strongly suggest that Stm1–80S represent a pool of dormant ribosomes that is most likely not readily split by the Dom34 system. Taken together, we conclude that Lso2 acts in protecting inactive 80S ribosomes as a hibernation factor, yet in contrast to Stm1, Lso2 stabilizes them in a rapid-to-recycle state for timely re-entrance into the active translation cycle.

So far, it has remained unclear what the actual signals for both Stm1 and Lso2 are in order to engage with idle 80S in their translational repressive subunit-clamping conformations. It is plausible to assume that both proteins can constitutively associate with ribosomes in an alternative permissive binding mode because Stm1, for example, has been observed in actively translating polysomes [57]. However, Stm1 was also suggested to trigger translation repression in yeast, perhaps by modulating access of translation elongation factor eEF3 [18,56], and to play a role in mRNA deadenylation [19]. Similarly, a role in early elongation was also attributed to Lso2 [21]. Thus, in the event of a starvation-driven translational shutdown, both potentially already ribosome (pre-) bound factors could quickly generate a pool of protected dormant 80S by adopting their repressive conformation on the same ribosome. Yet, the molecular signals required for either generation or disassembly of these distinct inactive 80S populations are not known. One possibility is that the relative population size of idle 80S ribosomes under specific stress conditions is already sufficient. On the other hand, Lso2–80S must be protected from recurrent recycling to avoid repetition of idle rounds of splitting and reformation of these 80S. Whether this is regulated by the energy levels in the cell or by more specific signals should be the subject of further investigation.

## Materials and methods

### Purification of recombinant Lso2

For overexpression of recombinant Lso2, the *LSO2* gene was cloned from yeast genomic DNA into a modified pRSF Duet1–SUMO vector downstream of the SUMO coding sequence. The resulting plasmid, coding for an SUMO–Lso2 fusion protein, was transfected into *Escherichia coli* (E. coli) BL21 (DE3) and grown in LB media supplemented with ampicillin. Cells were inoculated at an $OD_{600}$ of 0.05 and grown at 37°C to mid-log phase, at which point protein expression was induced by the addition of IPTG. Cells were harvested after 2 h of protein expression at 37°C by centrifugation at 3,500 × g for 10 min. Cell pellets were resuspended in lysis buffer (50 mM Tris-HCl [pH 8.0], 300 mM NaCl, 2 mM β-mercaptoethanol [β-ME]) and lysed using Microfluidics M-110L microfluidizer (Hyland Scientific, Stanwood, WA, USA). The membrane fraction was removed by centrifugation at 34,000 × g for 45 min. Clarified lysates were then loaded onto nickel resin equilibrated with 5 column volumes (CVs) of wash buffer (30 mM Tris-HCl [pH 8.0], 300 mM NaCl, 20 mM imidazole, 2 mM β-ME). Lso2 was cleaved in batch mode by addition of deubiquitin protease Ulp1 over night at 4°C and eluted in wash buffer lacking imidazole. The eluate was concentrated to 1 mL and applied on a Superdex 75 gel filtration column (Sigma-Aldrich, St. Louis, MO, USA) in 20 mM Tris-HCl (pH 8.0), 150 mM NaCl, 2 mM β-ME, yielding 1.1 mg purified Lso2 from 0.5 L culture at a concentration of 3 mg/ml.

## Purification of ribosomal subunits

Clarified cytoplasmic lysates obtained from isogenic *S.c.* S288C cells (MATα, HIS3, LEU2, ura3–52, TRP1, GAL2) were spun through a sucrose cushion (1 M sucrose, 30 mM Tris-HCl [pH 7.0], 500 mM KOAc, 25 mM Mg(OAc)$_2$, 5 mM β-ME, 0.1% Nikkol, 10 μg/mL cycloheximide) at 290,000 × *g* for 45 min. The ribosomal pellet was resuspended in buffer (50 mM Tris-HCl [pH 7.4], 500 mM KOAc, 2 mM MgCl$_2$, 2 mM DTT) and treated with puromycin (1 mM final concentration) for 15 min on ice and 10 min at 37˚C. Samples were then loaded onto 10%–40% sucrose gradients (50 mM HEPES [pH 7.4], 500 mM KOAc, 5 mM MgCl$_2$, 0.1 mM EGTA, 2 mM DTT) and centrifuged for 3 h at 284,000 × *g*. Gradient fractionation was carried out, and 40/60S subunit fractions were pooled and concentrated to 0.5 mL in grid buffer (20 mM HEPES [pH 7.4], 100 mM KOAc, 2.5 mM Mg(OAc)$_2$, 250 mM sucrose, 2 mM DTT).

## Reconstitution of the Lso2–80S complex

Purified ribosomal subunits were mixed at a 1:1 molar ratio and incubated under reassociation conditions in grid buffer containing 10 mM Mg(OAc)$_2$, 0.1% Nikkol for 10 min. Afterwards, a 10-fold molar excess of purified Lso2 was added 10 min prior to blotting.

## Native Lso2–80S complexes from *S.c.*

We identified Lso2-containing ribosomes by cryo-EM in several samples, in which ribosomal complexes were purified from yeast cells grown in minimal medium and overexpressing different target proteins on plasmids. The sample analyzed here was initially targeted at obtaining Upf1-containing ribosomal complexes from BY4741 yeast cells (*MATα; ura3Δ0; leu2Δ0; his3Δ1; met15Δ0; YMR080c::kanMX4*) harboring the plasmids pKB510, overexpressing a non-sense-mediated mRNA decay reporter, and pKB607, overexpressing a FLAG-tagged ATP-hydrolysis–deficient Upf1 mutant. Cryo-EM analysis of the elution fractions revealed a vast majority of idle 80S ribosomes lacking any density for Upf1 but instead revealing a subclass of Lso2-bound ribosomes.

In detail, cells were grown in minimal medium (yeast nitrogen base; −Leu −Ura dropout medium and 2% glucose) at 30˚C to an OD$_{600}$ of about 0.75. Cells were lysed using a freezer mill (SPEX 6970/EFM; SPEX, Metuchen, NJ, USA) before being resuspended in lysis buffer (20 mM HEPES [pH 7.4], 100 mM KOAc, 10 mM MgCl$_2$, 0.5% Triton X-100, 1:1,000 protease inhibitor pill [Roche: 04-693-132-001; Basel, Switzerland]). For the preparation, 40 g of lysed cell powder was used, and a cytosolic S100 fraction was prepared. First, membrane fractions were removed by centrifugation at 28,714 × *g* for 15 min, then cytosolic fractions were clarified by centrifugation at 92,387 × *g* for 20 min. The "S100" was added on 300 μl of magnetic FLAG beads (Sigma-M8823; Sigma-Aldrich) equilibrated with lysis buffer and incubated for 2 h at 4˚C. After washing 3 times with lysis buffer lacking Triton X-100, the sample was eluted with FLAG peptide (Sigma F4799; Sigma-Aldrich). Ribosomal eluate was spun through a 750 mM sucrose cushion prepared in elution buffer for 45 min at 290,000 × *g*. Pellets were resuspended in elution buffer and adjusted to a final concentration of approximately 4 $A_{260}$ ml$^{-1}$ for cryo-EM sample preparation. As mentioned above, only Lso2 could be visualized in this sample as additional ribosome binder. Similar observations were made when using the same protocol to obtain ribosomal complexes with other tagged proteins, indicating that the presence of Lso2 on vacant ribosomes is independent of the nature of the tagged bait protein.

## Native human hibernation complexes

Five × 15 cm plates of HEK293 Flp-In T-REx cells (Thermo Fisher Scientific, R78007; Waltham, MA, USA) at 80% confluency were harvested in 15 mL DMEM media by trypsinization and pelleted by centrifugation for 10 min at $150 \times g$. Cells were washed with ice-cold PBS and pelleted again before resuspending in 0.75 mL lysis buffer (20 mM HEPES/NaOH [pH 7.4], 100 mM KOAc, 10 mM Mg(OAc)$_2$, 125 mM sucrose, 1 mM DTT, 0.5 mM PMSF, 0.5% IGEPAL, protease inhibitor pill). Cells were incubated for 30 min at 4°C before pelleting the cell debris for 15 min at $21,000 \times g$. The clarified lysate was distributed on top of 10%–50% sucrose gradients prepared with lysis buffer lacking IGEPAL and centrifuged for 3 h at $284,000 \times g$. 80S fractions were collected, combined, and pelleted through a 1 M sucrose cushion of lysis buffer lacking IGEPAL by centrifugation for 1 h at $540,000 \times g$. Pellets were resuspended in 20 μL lysis buffer lacking IGEPAL, resulting in a final concentration of 130 $A_{260}$ ml$^{-1}$. For the preparation of cryo grids, the concentration was adjusted to 4 $A_{260}$ ml$^{-1}$.

## Purification of idle 80S ribosomes with Stm1

Wild-type BY4741 *S.c.* cells were prepared exactly as described previously [17]. In short, cells were grown to mid-log phase in YPD before pelleting at 30°C and being incubated in YP for a further 10 min at 30°C. Cells were pelleted and washed 3 times in wash buffer (30 mM HEPES [pH 7.4], 50 mM KCl, 2.5 mM Mg(OAc)$_2$, 0.5 mM EDTA, 2 mM DTT, protease inhibitor pill, 10% glycerol). After washing, cells were flash frozen in liquid nitrogen and lysed using a freezer mill (SPEX 6970/EFM), and lysed cell powder was stored at −80°C. Clarified lysates resuspended in wash buffer were loaded onto 10%–50% sucrose gradients in wash buffer lacking glycerol. After gradient fractionation, 80S peaks were pelleted through a 1-M sucrose cushion prepared in wash buffer, by centrifugation at $417,000 \times g$ for 45 min. Finally, purified ribosomal pellets were resuspended in storage buffer (20 mM HEPES [pH 7.5], 100 mM KOAc, 5 mM Mg(OAc)$_2$, 1 mM DTT). Aliquots were flash frozen and stored at −80°C.

## Purification of puromycin-treated 80S ribosomes

*S.c.* BY4741 cells were grown to mid-log phase in YPD at 30°C and harvested at a final OD$_{600}$ of 2.5. Cells were pelleted and washed once with water and once with 1% KCl before being resuspended in 100 mM Tris-HCl (pH 8), 10 mM DTT and incubated at room temperature for 15 min. After a final pelleting step, cells were resuspended in lysis buffer (10 mM HEPES [pH 7.5], 100 mM KOAc, 7.5 mM Mg(OAc)$_2$, 125 mM sucrose, 1 mM DTT, 0.5 mM PMSF, protease inhibitor pill) before being lysed using Microfluidics M-110L microfluidizer. Lysates were clarified via centrifugation at 4°C, first at $27,000 \times g$ for 15 min and again at $150,000 \times g$ for 30 min. Ribosomal fractions were then isolated by centrifugation through a double layer 1.5 M/2 M sucrose cushion (20 mM HEPES [pH 7.5], 500 mM KOAc, 5 mM Mg(OAc)$_2$, 1 mM DTT, 0.5 mM PMSF) at $246,500 \times g$ for 21 h. Supernatant fractions were discarded, and ribosomal pellets were resuspended in nuclease-free water. Ribosomes were mixed 1:1 with 2× puromycin buffer (40 mM HEPES [pH 7.5], 1 M KOAc, 25 mM Mg(OAc)$_2$, 2 mM puromycin, 2 mM DTT, and Amicon anti-RNase [AM2692; Sigma-Aldrich]) and incubated for 30 min at room temperature. Puromycin-treated ribosomes were loaded onto 10%–40% sucrose gradients in buffer conditions matching the previous sucrose cushion and subjected to ultracentrifugation at 4°C, $21,000 \times g$ for 20 h. 80S fractions were manually harvested from the gradients by monitoring absorption at 260 nm, and ribosomes were pelleted at $417,000 \times g$ at 4°C for 45 min. Finally, puromycin-treated 80S pellets were resuspended in storage buffer (20 mM HEPES [pH 7.5], 100 mM KOAc, 5 mM Mg(OAc)$_2$, 1 mM DTT). Aliquots were flash frozen and stored at −80°C.

## Purification of splitting factors

*S.c.* Dom34 was expressed with a C-terminal His tag in pET21a(+) vector and was purified from Rosetta2(DE3) *E. coli* with Ni-NTA affinity chromatography as previously reported [45].

*S.c.* Hbs1 was expressed with an N-terminal His tag in pET28b(+) vector and was purified from Rosetta2(DE3) *E. coli* with Ni-NTA affinity chromatography as previously described [45]. Both proteins, Dom34 and Hbs1, were purified in a final buffer of 20 mM Tris (pH 7.5), 200 mM NaCl, 5 mM β-ME, 0.1 mM PMSF, and aliquots were flash frozen and stored at −80˚C.

*S.c.* Rli1p (ABCE1) in high-copy pYES2Rli1 vector was overexpressed in *S.c.* strain WCGα. Cells were grown in YP −ura, 2% galactose, 1% raffinose media at 30˚C to mid-log phase and were harvested at a final $OD_{600}$ of 1.0. Cell pellets were washed with water before being flash frozen and stored at −80˚C. For lysis, cell pellets were thawed and washed once with 1% KCl for cell wall destabilization before being resuspended in 100 mM Tris (pH 8.0), 14 mM β-ME and incubated at room temperature for 15 min. Subsequently, cells were pelleted and resuspended in lysis buffer (75 mM HEPES [pH 8.0], 300 mM NaCl, 5 mM β-ME, 1% Tween, 20 mM imidazole, 2 mM $MgCl_2$, 10% glycerol) and lysed using Microfluidics M-110L microfluidizer. Lysates were clarified by centrifugation at $25,000 \times g$ for 10 min and filtered through a 0.45-μm filter before being loaded onto a HisTrap-HP 5mL affinity column using the ÄKTA pure system (Cytiva, Marlborough, MA, USA). The column was washed with 8 CVs wash buffer (50 mM HEPES [pH 8.0], 500 mM NaCl, 5 mM β-ME, 20 mM imidazole, 2 mM $MgCl_2$, 10% glycerol) before eluting with 8 CVs over a 0%–100% gradient from wash to elution buffer (wash buffer with 300 mM imidazole). Peak fractions were concentrated to 1 mL before being loaded onto Superdex 200 (Sigma-Aldrich) for size exclusion chromatography. Aliquots of pure ABCE1 in 20 mM Tris (pH 7.5), 200 mM NaCl, 5 mM β-ME, and 5% glycerol were flash frozen and stored at −80˚C.

*S.c.* eIF6 was cloned into p7XC3GH (Addgene #47066; Watertown, MA, USA) fused at the C-terminus to a 3C protease cleavage site, GFP, and 10× His. The plasmid was transformed into *E. coli* BL21 (DE3), which was grown at 37˚C to mid-log phase. The temperature was reduced to 16˚C, and protein overexpression was induced with IPTG for overnight expression. Cells were harvested ($4,400 \times g$, 4˚C, 8 min), washed with phosphate-buffered saline, and resuspended in lysis buffer (20 mM Tris-HCl [pH 8.0], 300 mM NaCl, 2 mM β-ME, and protease inhibitor pill) before being lysed with Microfluidics M-110L microfluidizer. Lysates were clarified by centrifugation at $30,596 \times g$ at 4˚C for 20 min. The supernatant was loaded onto TALON metal affinity resin (Takara Bio, Mountain View, CA, USA) equilibrated in lysis buffer and incubated on a rotating wheel at 4˚C for 40 min. Flowthrough was collected, and resin was washed 3 times with lysis buffer + 10 mM imidazole before being incubated with elution buffer (lysis buffer, 10 mM imidazole, 0.25 mg/mL 3C protease) for 30 min at 4˚C. Eluted protein was concentrated to 1 mL before being loaded onto Superdex 200 (Sigma-Aldrich) for size exclusion chromatography in the final buffer (50 mM HEPES [pH 7.5], 500 mM KCl, 2 mM $MgCl_2$, 2 mM β-ME). Aliquots were flash frozen and stored at −80˚C.

## Splitting assay

Splitting assays were assembled in 50-μL reactions in a final SA buffer (20 mM Tris [pH 7.5], 100 mM KOAc, 4 mM $Mg(OAc)_2$, 5 mM β-ME), with 5 pmol purified ribosomes (Stm1-bound, Lso2-bound, or puromycin-treated) and a 5× molar excess of each factor included in the reaction. Control reactions were assembled with ribosomes and eIF6. Splitting reactions included ribosomes eIF6, 1 mM ATP, 1 mM GTP, and splitting factors (Dom34, Hbs1, and ABCE1). Reactions were incubated on ice for 30 min before being loaded onto

10%–50% sucrose SA buffer gradients and subjected to ultracentrifugation for 3 h at $284,000 \times g$. Sucrose density gradients were subjected to mechanical fractionation and UV spectroscopy. Quantification of peaks was performed by estimating the integral using the trapezoid rule [58]. Let $(x_n \mid A_n)$ be the data points recorded, with x being the distance along the gradient and A the absorption at 260 nm. The area $S_{ab}$ under one peak from $x_a$ to $x_b$ was approximated as

$$S_{ab} = \sum\nolimits_{n=a}^{b-1} 0.5 \left(A_n + A_{n+1}\right) \left(x_{n+1} - x_n\right)$$

Relative splitting efficiencies were calculated as ratios of peak areas and averaged across experiments. Errors shown in normalized results were estimated assuming linear propagation of statistical uncertainties.

## Cryo-EM analysis

For all yeast samples, approximately 2.5–6 $A_{260}$ ml$^{-1}$ of ribosome were applied to 2-nm precoated Quantifoil R3/3 holey carbon support grids. Data were collected on a Titan Krios TEM (Thermo Fisher Scientifc) equipped with a Falcon II direct detector at 300 keV under low-dose conditions using approximately 28 electrons per Å$^2$ for 10 frames in total (S1 Table). The defocus range was between −1.1 to −2.3 μm, and for semiautomated data acquisition, the software EM-TOOLS (TVIPS) was used. Magnification settings resulted in a pixel size of 1.084 Å pixel$^{--1}$. Original image stacks were summed and corrected for drift- and beam-induced motion at the micrograph level using MotionCorr2 [59]. The estimation of contrast transfer function (CTF) and the resolution range of each micrograph was performed with Gctf [60].

For the human sample, 4 $A_{260}$ ml$^{-1}$ of ribosomes were applied to 2-nm precoated Quantifoil R3/3 holey carbon support grids. Data were collected on a Titan Krios TEM (Thermo Fisher Scientific) equipped with a Falcon III direct detector at 300 keV (S1 Table). The defocus range was between −0.8 to −3.2 μm, and for semiautomated data acquisition, the software EPU (Thermo Fisher Scientific) was used. Frame alignment was performed using MotionCorr2 [59]. The estimation of CTF was performed with Gctf [60].

## Structure of the in vitro-reconstituted yeast Lso2–80S complex

After manual screening for ice quality, 10,313 micrographs were used for automated particle picking in Gautomatch (https://www2.mrc-lmb.cam.ac.uk/research/locally-developed-software/zhang-software/), yielding 1,413,783 initial particles. Upon 2D classification in RELION 3.0, 781,257 particles were selected for a consensus 3D refinement. After 3D classification, an Lso2-containing class (88,523 particles) was obtained and refined, including CTF refinement, to an average resolution of 3.4 Å with local resolution ranging from 3–7 Å (3.2–4.5 for Lso2). All other classes contained only 80S ribosomes with no additional factors, tRNA, or mRNA. A classification scheme is displayed in S1A Fig.

## Structure of the native yeast Lso2–80S complex

After manual screening for ice quality, 8,600 micrographs were used for automated particle picking in Gautomatch, yielding 585,801 initial particles. Upon 2D classification in RELION 3.0, 486,383 particles were selected for a consensus 3D refinement. Two rounds of 3D classification and 3D refinement resulted in reconstructions of a low-resolution Lso2–80S complex from 29,735 particles. This data set was later merged with a subsequent data set, wherein, 8,400 micrographs underwent automated particle picking in Gautomatch, yielding 381,233 initial particles. After 2D classification in RELION 3.0, 163,303 particles were selected for a consensus 3D refinement. 3D classification and 3D refinement resulted in reconstructions of a low-

resolution Lso2–80S complex from 24,085 particles. The resulting merged data set of 53,820 particles underwent a consensus 3D refinement before focused sorting on the intersubunit space (A- and P-site tRNA positions) resulting in one tRNA containing class of 18,869 particles and one Lso2-containing class of 34,951 particles. The Lso2-containing class underwent 1 final round of 3D refinement and CTF refinement, resulting in an Lso2–80S complex reconstruction at 3.5 Å. Discarded classes contained 80S ribosomes with no additional factors, tRNA, or mRNA. A classification scheme is displayed in S1B Fig.

## Structure of the human hibernation complex

After manual screening for ice quality, 6,145 micrographs were used for automated particle picking in Gautomatch, yielding 332,890 initial particles. Upon 2D classification in RELION 3.0, 156,750 particles were selected for a consensus 3D refinement. Extensive 3D classification followed by 3D refinement and CTF refinement resulted in reconstructions of an SERBP1–eEF2–80S complex, a CCDC124–80S complex, and an EBP1–80S complex at 3.1 Å, 3.0 Å, and 2.9 Å, respectively. A classification scheme is displayed in S4 Fig.

## Molecular models of yeast and human hibernating ribosomes

The crystal structure of the *S.c.* (Protein Data Bank [PDB]: 5NDG) the human cryo-EM structures (PDB: 6EK0 and 4V6X) [20], and the human EBP1 crystal structure (PDB: 2Q8K) [36] were used as initial models to build the 80S ribosomes and EBP1, respectively. In general, the ribosome/EBP1 models were rigid-body fitted into our cryo-EM maps in Chimera [61], followed by manually adjusting in Coot according to the densities [62].

Because of the flexibility, the C-termini of both Lso2 and CCDC124 and the N-terminus of CCDC124 are missing in our final model, but all the other regions were de novo built in Coot. A homology model of the human eEF2 was generated using Swiss-Model server [63] based on the *Sus scrofa* model (PDB: 3J7P) [64]. The human SERBP1 model was adjusted from human ribosome structure (PDB: 4V6X) [20].

All the final models (S2, S5 Figs and S1 Table) were real-space refined with secondary structure restraints using the PHENIX suite [65], and the final model evaluation was performed with MolProbity [66]. Maps and models were visualized and figures created with the PyMOL Molecular Graphics System (Version 1.7.4, Schrödinger, LLC) and ChimeraX [67].

Standard model-to-map validations were performed according to [68] to ensure that models are not overfitted.

## Supporting information

**S1 Fig. 3D classification scheme and local resolution of Lso2–80S reconstructions.** (A–B) Sorting schemes for reconstituted (A) and native (B) Lso2–80S reconstructions. For the native Lso2–80S structure, particles containing Lso2 were merged from 2 individual collections after 3D classification because of relatively low occupancy with Lso2 (5% and 16% of all particles, respectively). For both reconstituted and native complexes, the Lso2-containing classes were locally classified using an ellipsoid mask covering the A- and P-site tRNA binding sites of the 80S. The Lso2-containing classes were refined to an overall resolution of 3.5 Å and 3.4 Å, respectively. (C) The local resolution range for 80S ribosome and isolated Lso2 is indicated. A, acceptor; Lso2, late-annotated short open reading frame 2; P, peptidyl.
(TIF)

**S2 Fig. Validation of the Lso2–80S model.** (A) View highlighting the model for the N-terminus of Lso2 interacting with rRNA helix h28 (G1150) and h45 (G1768) fitting into respective

densities (transparent gray mesh). (B) View focusing on the Lso2 N-terminal R4 interacting with h44. (C) View focusing on the stacking of Lso2 hinge residue W42 inside a cleft formed by large subunit proteins eL42 and uL5. (D) View focusing on interactions between the hinge region of Lso2 (R47) and uL5 residues. (E) Fit of the entire Lso2 model into isolated density. (F) Overall FSC curves, map to model FSC curves, and half map to model FSC curves for validation of the Lso2–80S model fitting the reconstituted and native Lso2–80S reconstructions. FSC, Fourier shell correlation; Lso2, late-annotated short open reading frame 2
(TIF)

**S3 Fig. CCDC124 and EBP1 are ribosome-associated in human cell lysates.** HEK293T cell lysates were fractionated through 10%–40% sucrose gradients. Fractions were collected, and western blotting using antibodies against CCDC124 and EBP1 was performed. Both factors were found to associate with 80S ribosomes. CCDC124, coiled-coil domain containing short open reading frame 124; EBP1, ErbB3-binding protein 1; HEK, human embryonic kidney.
(TIF)

**S4 Fig. 3D classification scheme and local resolution for native human idle ribosomes.** (A) In a first round of 3D classification, particles were separated, containing E-site tRNA and EBP1 and either CCDC124 (in the nonrotated state) or SERBP1 and eEF2 (in the rotated-2) state. Independent subclassifications using a regional mask were then performed using ellipsoid masks covering the A- and P-site tRNA binding sites (for CCDC124 and eEF2/SERBP1, respectively) or on the peptide exit site (for EBP1). This yielded homogenous ribosome classes containing exclusively either CCDC124 and EBP1 or SERBP1/eEF2 and EBP1. The classes displaying CCDC124–80S, SERBP1/eEF2–80S and EBP1-enriched SERBP1/eEF2–80S were refined to overall resolution of 3.0 Å, 3.1 Å, and 2.9 Å. (B–D) Local resolution ranges for CCDC124–80S (B), EBP1–80S (C), and eEF2/SERPB1–80S (D) and isolated CCDC124 and EBP1 are indicated. A, acceptor; CCDC124, coiled-coil domain containing short open reading frame 124; EBP1, ErbB3-binding protein 1; eEF2, eukaryotic elongation factor 2; P, peptidyl; SERBP1, SERPINE1 mRNA-binding protein 1
(TIFF)

**S5 Fig. Validation of the human CCDC124–80S, SERBP1–eEF2–80S, and EBP1–80S models.** (A–B) View highlighting the hinge region of CCDC124 interacting with H85 (A) and H84 (B) of the 28S rRNA. (C) View focusing on the interaction of CCDC124 W39 stacking with residues of uL5 in a manner distinct from Lso2, wherein eL42 also participates in stabilizing Lso2 W42. (D) Fits of the entire CCDC124 and EBP1 models into the respective isolated densities. (E) Overall FSC curves, map to model FSC curves, and half map to model FSC curves for validation of the final CCDC124–80S, eEF2/SERPB1–80S, and EBP1–80S models fitting the respective densities. CCDC124, coiled-coil domain containing short open reading frame 124; EBP1, ErbB3-binding protein 1; FSC, Fourier shell correlation; Lso2, late-annotated short open reading frame 2; SERPB1, SERPINE1 mRNA-binding protein 1
(TIF)

**S6 Fig. Binding of EBP1 near tunnel exit on the 60S.** (A) Close-up view showing the overall positioning of EBP1 at the peptide exit site. (B) Zoom on the interaction between EBP1 and H59. (C) Comparison of the EBP1 position with other exit-site ligands. EBP1 binding would overlap with a majority of nascent chain-interacting factors such as NatA [43], RAC [40], SRP [69], or Sec61 [41,42,70]. EBP1, ErbB3-binding protein 1; NatA, N-α-Acetyltransferase; RAC, ribosome associated complex; Sec61, secretory protein 61; SRP, signal recognition particle.
(TIF)

**S7 Fig. Overlay of Dom34 with 18S rRNA in rotated-2 and nonrotated states.** Dom34 in complex with Hbs1 (and also ABCE1) is usually found in the nonrotated state. The structure of a yeast 80S in the nonrotated state (PDB: 6Q8Y; representing Lso2–80S) [25] was superimposed with one in the rotated-2 state (PDB: 5JUP; representing Stm1–80S) [26] based on the 60S subunits. We note that the Dom34 N-terminal domain (taken from PDB: 3IZQ) [45] would clash with 18S rRNA helix h18 and h34 in the rotated-2 state. This indicates that the rotated (Stm1-containing) 80S ribosome cannot be split by the Dom34 splitting system. Dom34, duplication of multilocus region 34; Hbs1, Hsp70 subfamily B suppressor; Lso2, late-annotated short open reading frame 2; PDB, Protein Data Bank; Stm1, suppressor of target of Myb protein 1
(TIF)

**S8 Fig. Sucrose gradient profiles for in vitro splitting assays.** UV profiles for splitting assays. Assays contain 5 pmol of ribosomes (treated with puromycin or bound to Lso2 or bound to Stm1) and 25 pmol of factors (60S subunit anti-association factor eIF6, Dom34, Hbs1, and ABCE1). 50-µl reactions were spun through a 10%–50% sucrose gradient, and an absorption profile at 260 nm ($A_{260}$) was recorded. All experiments were carried out in triplicates (as indicated in different shades of green). Raw data can be found in S1 Data. Dom34, duplication of multilocus region 34; eIF6, eukaryotic initiation factor 6; Hbs1, Hsp70 subfamily B suppressor; Lso2, late-annotated short open reading frame 2; Stm1, suppressor of target of Myb protein 1
(TIF)

**S9 Fig. Quantification of in vitro splitting assays.** The relative abundance of 80S ribosomes and subunits in in vitro splitting assays was estimated by comparing the areas under $A_{260}$ absorption peaks from 40S, 60S, and 80S subunits (calculated as described in Methods). Note that Stm1–80S have the highest relative abundance after splitting, whereas Lso2–80S show the lowest relative abundance after splitting. Calculations can be found in S2 Data. Lso2, late-annotated short open reading frame 2; Stm1, suppressor of target of Myb protein 1
(TIF)

**S10 Fig. Component requirements for splitting assay.** (A) In vitro splitting assays were assembled (materials and methods) with 80S ribosomes reconstituted with Lso2, omitting individual factors or nucleotides as annotated. UV profiles for splitting assays were collected following separation over a 10%–50% sucrose gradient; absorption profiles were collected at 260 nm. Splitting was observed in reactions when all components (eIF6, ABCE1, Hbs1, Dom34, 1 mM ATP, 1 mM GTP) were present or in the absence of Hbs1. Conversely, splitting was not observed when Dom34, Hbs1, or nucleotides were omitted from the reaction or in reactions containing eIF6 and nucleotides without other factors. (B) Percent splitting was calculated by comparing the areas under $A_{260}$ absorption peaks from 40S, 60S, and 80S subunits (calculated as described in Methods) and comparing the relative abundance of 80S ribosomes and subunits. Raw data can be found in S3 Data. Dom34, duplication of multilocus region 34; eIF6, eukaryotic initiation factor 6; Hbs1, Hsp70 subfamily B suppressor; Lso2, late-annotated short open reading frame 2
(TIF)

**S11 Fig. 3D classification for Stm1–80S ribosomes.** Stm1–80S were prepared following protocols previously reported [17] to enrich Stm1 binding and used in in vitro splitting reactions. Stm1–80S were collected from the sucrose gradient following in vitro splitting reactions with and without splitting factors (control). These 80S fractions were analyzed by cryo-EM in which particles were assessed for ribosome rotational states and presence of Stm1. (A) 80S ribosome population from the control splitting assay. (B) 80S ribosome population remaining

following incubation with splitting factors. An initial 3D classification revealed that a vast majority of 80S are in the rotated state. These particles were further subjected to local classification using and ellipsoid mask covering the region of the mRNA channel of the 40S to classify for Stm1-containing particles (approximately 50%), which were further refined. Stm1 density is displayed in orange. cryo-EM, cryo-electron microscopy; Stm1, suppressor of target of Myb protein 1
(TIF)

**S1 Table. Cryo-EM data collection, refinement, and validation statistics.** Overview of cryo-EM data collection, data processing, and model-fitting parameters for yeast native and reconstituted Lso2–80S complexes and for human CCDC124–EBP1 and SERBP1–eEF2–EBP1 complexes. CCDC124, coiled-coil domain containing short open reading frame 124; cryo-EM, cryo-electron microscopy; EBP1, ErbB3-binding protein 1; eEF2, eukaryotic elongation factor 2; Lso2, late-annotated short open reading frame 2; SERBP1, SERPINE1 mRNA-binding protein 1
(DOCX)

**S1 Data. UV absorption profiles related to figures Figs 5C–5D and S8.** This file contains raw data from the UV absorption profiles for each splitting assay gradient in sheets 2–19. Splitting assays were carried out in triplicate, for each population, for both experimental and negative control reactions. Sheets 20–25 contain aligned data, including baseline and peak alignment from triplicate experiments, which are the basis for the curves shown in S8 Fig. For Fig 5C and 5D, only one of these triplicates is shown. Charts 1–6 show the plots for each triplicate experiment.
(XLSX)

**S2 Data. Calculations related to figures Figs 5E and S9.** Determination of the peak area and error bar calculation. From the UV absorption profile of each gradient, areas under each peak corresponding to ribosomal subunits or 80S ribosomes were measured. To this end, beginning and end of each peak were determined from the profile visually, and the area under the curve in between these 2 points was approximated from individual data points using the trapezoid rule. For each triplicate experiment, the fraction of total peak area (40S + 60S + 80S) taken up by subunit peaks (40S + 60S) was averaged. From this, the fold change in this fraction between experiments using eIF6 only and those using the full complement of splitting factors was calculated. These values were then normalized to the control experiment using puromycin-treated ribosomes. Error bars were calculated using Gaussian error propagation for the (uncorrelated) uncertainties of the averaged peak areas. eIF6, eukaryotic initiation factor 6
(XLSX)

**S3 Data. UV absorption profiles and calculations related to S10 Fig.** This file contains raw data from the UV absorption profiles for each splitting assay control experiment shown in S10 Fig. Additional calculations used for defining peak boundaries and quantification summary can be found in S9 Data and the corresponding legend.
(XLSX)

## Acknowledgments

We thank S. Rieder, C. Ungewickell, A. Gilmozzi, J. Musial, and H. Sieber for excellent technical assistance.

## Author Contributions

**Conceptualization:** Jennifer N. Wells, Wendy Gilbert, Jingdong Cheng, Roland Beckmann.

**Data curation:** Jennifer N. Wells, Robert Buschauer, Timur Mackens-Kiani, Katharina Best, Hanna Kratzat, Otto Berninghausen, Roland Beckmann.

**Formal analysis:** Jennifer N. Wells, Robert Buschauer, Thomas Becker, Jingdong Cheng, Roland Beckmann.

**Funding acquisition:** Roland Beckmann.

**Investigation:** Jennifer N. Wells, Robert Buschauer, Wendy Gilbert, Roland Beckmann.

**Methodology:** Jennifer N. Wells, Timur Mackens-Kiani, Otto Berninghausen, Jingdong Cheng, Roland Beckmann.

**Project administration:** Jennifer N. Wells, Thomas Becker, Roland Beckmann.

**Resources:** Robert Buschauer, Otto Berninghausen, Thomas Becker, Roland Beckmann.

**Software:** Robert Buschauer, Otto Berninghausen, Roland Beckmann.

**Supervision:** Thomas Becker, Jingdong Cheng, Roland Beckmann.

**Validation:** Jennifer N. Wells, Robert Buschauer, Thomas Becker, Jingdong Cheng, Roland Beckmann.

**Visualization:** Jennifer N. Wells, Robert Buschauer, Timur Mackens-Kiani, Jingdong Cheng, Roland Beckmann.

**Writing – original draft:** Jennifer N. Wells, Robert Buschauer, Thomas Becker, Jingdong Cheng, Roland Beckmann.

**Writing – review & editing:** Jennifer N. Wells, Robert Buschauer, Timur Mackens-Kiani, Katharina Best, Hanna Kratzat, Thomas Becker, Wendy Gilbert, Jingdong Cheng, Roland Beckmann.

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
