## [Editor Report · Decision Letter 0]

27 Apr 2020

Dear Dr Beckmann, 

Thank you for submitting your manuscript entitled "Structure and function of yeast Lso2 and human CCDC124 bound to hibernating ribosomes" for consideration as a Research Article by PLOS Biology.

Your manuscript has now been evaluated by the PLOS Biology editorial staff, as well as by an academic editor with relevant expertise, and I'm writing to let you know that we would like to send your submission out for external peer review.

Please re-submit your manuscript within two working days, i.e. by Apr 29 2020 11:59PM.

Kind regards,

Roli Roberts

Senior Editor

PLOS Biology

---

## [Decision Letter · Decision Letter 1]

23 May 2020

Dear Dr Beckmann,

Thank you very much for submitting your manuscript "Structure and function of yeast Lso2 and human CCDC124 bound to hibernating ribosomes" for consideration as a Research Article by PLOS Biology. As with all papers reviewed by the journal, yours was evaluated by the PLOS Biology editors as well as by an Academic Editor with relevant expertise and in this case by four independent reviewers. 

You'll see that all four reviewers are very positive about your study, and make only minor requests for improvements. Based on the reviews, we will probably accept this manuscript for publication, assuming that you will modify the manuscript to address the remaining points raised by the reviewers. Please also make sure to address the data and other policy-related requests noted at the end of this email.

We expect to receive your revised manuscript within two weeks. Your revisions should address the specific points made by each reviewer. In addition to the remaining revisions and before we will be able to formally accept your manuscript and consider it "in press", we also need to ensure that your article conforms to our guidelines. A member of our team will be in touch shortly with a set of requests. As we can't proceed until these requirements are met, your swift response will help prevent delays to publication.

*Copyediting*

*Published Peer Review History*

*Early Version*

*Submitting Your Revision*

Sincerely,

Roli Roberts

Senior Editor

PLOS Biology

DATA POLICY:

Please ensure that your structural data are deposited appropriately (PDB, EMDB, etc). In addition, we ask that all individual quantitative observations that underlie the data summarized in the figures and results of your paper be made available in one of the following forms:

Regardless of the method selected, please ensure that you provide the individual numerical values that underlie the summary data displayed in the following figure panels as they are essential for readers to assess your analysis and to reproduce it: Figs 5E, S9, and S10B. NOTE: the numerical data provided should include all replicates AND the way in which the plotted mean and errors were derived (it should not present only the mean/average values).

REVIEWERS' COMMENTS:

Reviewer #1:

The study by Wells and Buschauer et al. from the Beckmann laboratory reports on the cryoEM structure-based discovery of conserved proteins that stabilize inactive or "hibernating" ribosomes from yeast to humans. These structures are important because they reveal, in molecular terms, how yeast and human cells adapt to fluctuating environmental nutrient availability by reversibly decreasing protein synthesis while simultaneously protecting and maintaining inactive ribosomes for future use. These new structures are insightful and help explain genetic and biochemical data from prior reports about Lso2/CCDC124. The structures also reconcile previous work on an alternative hibernation factor, Stm1/SERBP1, by defining the different ribosome conformations that Lso2/CCDC124 versus SERBP1 stabilize. Biochemical data, including mass spectrometry, confirm the co-purification of CCDC124 and EBP1 with hibernating ribosomes. The structural finding that EBP1, moreover, occludes the surface of the 60S near the exit tunnel is novel and will serve as a useful reference for future functional studies. The structural work is outstanding and thoroughly described. My only question concerns the ribosome-splitting assay and its interpretation. 

Major comments

1. The authors predicted that Pelota/Hbs1/ABCE1 will dissociate hibernating ribosomes that are stabilized by Lso2/CCDC124. Specifically, they claim that Pelota/Hbs1/ABCE1 splitting will recycle Lso2/CCDC124-protected ribosomes—but not SERBP1-protected ribosomes—because of the different conformational states seen in their structures. To test whether Lso2/CCDC124 stabilize non-rotated 80S ribosome in a splitting-competent, and therefore recycling-competent state, the authors compare puromycin splitting (control) with Pelota/Hbs1/ABCE1 splitting of Lso2- versus Stm1/SERBP1-engaged 80S ribosomes. By employing a series of controls (omitting nucleotides, omitting individual splitting factors), the authors observe that Lso2-reconstituted ribosomes are split efficiently in vitro, while Stm1-reconstituted 80S ribosomes are not. There is a fly in the ointment though, because "splitting was observed in reactions when all components (eIF6, ABCE1, Hbs1, Dom34, 1mM ATP, 1mM GTP) were present, or in the absence of Hbs1." Shouldn't the absence of Hbs1 have prevented splitting in this negative control?

2. The comparison with Stm1-containing yeast 80S ribosomes obtained after glucose starvation is valuable, but the comparison does introduce confounding variables because these Stm1-80S particles were obtained from glucose starved yeast rather than reconstituted from subunits and pure Lso1. The authors may want to emphasize that these ribosomes are an imperfect control because they differ from the Lso2-reconstituted ribosomes in many ways, including unknown co-purifying factors or unknown modifications, etc. So although the Stm1-80S complexes were found to be "essentially resistant to splitting by the Dom34 system" this may be due to other differences in the Stm1 particle, not just its rotation status. 

Reviewer #2:

This manuscript reports structural and biochemical evidence that Lso2 stabilizes an inactive ribosome confirmation poised for reactivation by ribosome rescue and recycling factors Dom34 and Rli1. Structures of inactive yeast ribosomes with Lso2 bound are presented, with reconstituted addition of purified Lso2 and native purification from growth-limited yeast appearing very similar. These ribosomes resemble a post-translocation, non-rotated conformation, with Lso2 occupying the peptidyl-tRNA site and interfering with tRNA or factor binding. A fraction of ribosomes from quiescent cultured human cells show a similar non-rotated quiescent state bound to Ccdc124. These human cells also contained inactive ribosomes bound by Serbp1 and eEF2. Further, both of these populations of inactive ribosomes were bound by Ebp1, which interacts near the exit tunnel of the ribosome at a similar site to many co-translational chaperones. The yeast orthologue of Ebp1, Arx1, has a well-supported role in large subunit biogenesis and nuclear export, and these data suggest functional differences between the yeast and human proteins.

The non-rotated conformation of Lso2 / Ccdc124-bound ribosomes and the partially accessible A site led to the hypothesis that these ribosomes might be poised for reactivation by Dom34*Hbs1. This complex, working in conjunction with the recycling factor Rli1, dissociates subunits of stalled ribosomes. Here, it is shown that Dom34 and Rli1 together split Lso2-bound non-rotated ribosomes, but not Stm1-bound rotated ribosomes. From these results, it is inferred that Lso2-bound ribosomes can be rapidly reactivated by splitting, offering an explanation for the importance of Dom34 in resuming growth from quiescence.

The high-quality structural and biochemical data here advance our understanding of ribosome storage and reactivation in eukaryotes. Further, this work points towards a possible functional significance for the Lso2-bound ribosome pool that is consistent with genetic evidence. I would support the publication of this manuscript in PLoS Biology, provided the following points are addressed:

1. Other recent studies and pre-prints analyze the structure of Ebs1 on human ribosomes and reach slightly differing conclusions. Wild et al., (Nat Commun 2020) present a structure that seems to share key features with the present study and suggest that Ebs1 blocks the exit tunnel. Kraushar et al. (bioRxiv 2020.02.08.939488) present structural evidence that Ebs1 associates with translating ribosomes. They also argue that Ebs1 is very high abundance at least in certain cell types, and might associate with a substantial fraction of all ribosomes, active or inactive. Ebs1 structures should be discussed in light of these other results, as should the compatibility of Ebs1 with active translation.

2. The data presented here do pose some questions relative to the genetic results regarding Dom34. The contrast between in vitro and in vivo requirements for Hbs1 are discussed in den Elzen et al. (ref 15 of this manuscript) and probably merit mention here. More puzzling, though, why does the loss of Stm1 reduce dependency on Dom34 activity, and overexpression of Stm1 increase dependency on Dom34 activity? 

3. As a minor point, p. 10 l. 183, "Two distinct, inactive ribosomal species are present IN eukaryotes"

Reviewer #3:

The mechanisms by which ribosomes are idled in the cell during periods of non-proliferation have been extensively examined in bacteria for multiple decades. Recently, high-resolution structural information has been revealed that shed light on the multiple distinct mechanisms for idling that exist by which the cessation of ribosome activity can be achieved. This includes a long-term storage mechanism that generates 100S "disosomes". Such mechanism play critical roles in bacterial physiology and fitness. 

The present investigations are very important as the add to the emerging understanding of ribosome-arrest and hibernation in eukaryotic (yeast and human) cells about which very little is known. The structural and biochemical investigations presented suggest that the complex of Lso1, a protein recently discovered in yeast, with the ribosome has the capacity to bind in a manner that may equips them to be recycled by the "splitting factor" Dom34. The authors attribute this capacity of Lso1s and the human homologue CCDC124's to stabilization of the 80S ribosome in an unrotated conformation, which is competent for Dom34 binding and activity. 

The authors elegantly substantiate the importance of this finding though parallel structural and biochemical investigations of arrested 80S ribosome complex that are bound to either protein factors, smt1 (yeast) or SERBP1 (human), which stabilize ribosomes in a rotated conformation, which is recognized and further stabilized by eEF2 and tRNA binding within the E site. In carefully executed, appropriately integrated structural and biochemical assays, the authors provide what appears to be concrete evidence that Dom34 is simply unable to engage the rotated ribosome conformation that is stabilized by stm1/SERBP1 even though there should be adequate space to do so in the absence of eEF2. These findings provide a compelling and complete story that reveals the existence of distinct types/classes of arrested 80S complexes and opens the door for numerous downstream investigations aimed at exploring the physiological roles of such complexes in distinct tissues and cellular environments and the regulatory mechanisms and timing that control the formation and disassembly of such arrested complexes. 

The manuscript is very clear and well written, the figures efficiently convey the findings and the data strongly support the major conclusions drawn. There are multiple high-resolution structures presented: some have not been observed previously; others are reported and higher-resolution that analogous complexes reported previously. For this reason I recommend publication.

The questions that were raised for me during review, which the authors may wish to address relates to the cellular concentrations of Lso1, stm1 and SERBP1 and how this may tie into the distinct regulatory mechanisms that they propose. Are these factors repressed under normal, proliferative conditions and somehow activated during stress? Or are they always present in the cell at relatively high concentrations? Based on my understanding of the authors discussion, Initiation is where they suggest Lso1/CCD124-mediated regulation manifests. However, the Dom34 studies suggest that Lso1/CCD124 functions may also manifest during translation elongation, correct? Impacts on elongation would require Lso1/CCD124 binding to actively translating ribosomes. Is this the correct inference?

Whether it be initiation or elongation (or both), the observed pose of Lso1/CCD124 on the ribosome also raises the question of how these proteins bind in the first place. The structures identifies that Lso1/CCD124 are positioned on the 80S ribosome in a manner that precludes tRNA binding at both the P and A sites. Importantly, one or both of these tRNA binding sites would largely be occupied during scanning initiation steps, 80S initiation complex formation and translation elongation. Does Lso1/CCD124 stimulate tRNA release (at least P site tRNA release)? 

It would seem prudent to speculate on this aspect of the proposed model in the discussion section as it would seem relevant to specifying which population of ribosomes in the cell are indeed targeted. 

Another curious observation made by the authors, which may be related somehow, relates to the observed binding of EBP1 to the exit tunnel of their arrested complexes. The observed occupancy levels in their samples, and the positioning of this protein at the exit tunnel so as to block the binding of chaperones and other active protein synthesis interacting proteins, would suggest that Lso1/CCD124 bind ribosomes that have somehow already terminated translation. In this model, both proteins may bind so as to ensure that they are efficiently halted to prevent further activity or that they are shunted to recycling to ensure downstream initiation events rather than being employed elsewhere or degraded. Either way (or other ways) some mention should be made about what triggers EBP1 binding to the arrested ribosomes. It may also be worth mentioning in this context the potential impacts of sequestering ES27 near the exit tunnel as Figure 4A would appear to suggest. Is EBP1 also present at high concentrations in the cytoplasm? Does EBP1 simply have the tendency to bind ribosomes that don't have other factors bound (ie. perhaps lacking nascent peptides). The authors state EBP1 a ribosome biogenesis factor, which would suggest nucleolar localization. Is there any information out there on its expression levels and/or cytoplasmic concentration?

To be clear, the inclusion of such details and speculations are just suggestions. If the authors wish to not speculate further, this should by no means prevent the authors from moving forward with reporting this exciting and new set of findings.

Reviewer #4:

This work by Wells focuses on the structural studies of the hibernating ribosomes from yeast and human. The authors determined several cryo-EM structures at a very good resolution of the yeast 80S ribosome with a new hibernation factor Lso2 and human 80S ribosome in complex with protein CCDC124, which is homologous to Lso2. Also, the authors determined a new structure of the human 80S ribosome in complex with alternative hibernation factor SERBP1. This work represents a long-awaited continuation of the previous work by the same and other groups and reveals many interesting structural aspects of ribosome hibernation in eukaryotes. Studying the mechanisms of ribosome hibernation and the protein factors mediating this process in eukaryotes is crucial not only from a fundamental point of view but also has far-reaching medical implications because many of these hibernation factors are related to cancer development.

One of the most interesting findings of this work is that eukaryotes exploit two distinct mechanisms for ribosome hibernation: one employs proteins Lso2/CCDC124 and preserves 80S ribosomes in the unrotated state, while the other utilizes proteins Stm1/SERBP1 and traps the ribosomes in the ratcheted state. This is quite distinct to the mechanisms of ribosome hibernation operating in bacteria. Also, this work highlights how distinct and diverse are the ribosome hibernating factors from pro- and eukaryotes. In addition to quite intense structural studies, the authors confirmed some of their ideas and hypotheses with biochemical in vitro experiments using purified Dom34 and other ribosome splitting factors. I find the results of the ribosome splitting experiments and subsequent structural analysis of some of the complexes as "a cherry on the cake" that definitely reinforces the whole story.

Overall this work is an excellent and exemplary research study that was accomplished in the best traditions of the modern cryo-EM structural biology! In my opinion, this work represents a significant conceptual advance in the field and also answers several long-standing questions. I think that the main findings of this study merit publication in PLOS Biology and are ideally suited for this journal in scope. Moreover, the manuscript is very well-written and organized. There are simple and bright illustrations that are self-explanatory. It appears to me that the text and figures have already been thoroughly revised, and it seems to be close to its energetic minimum. In conclusion, I am enthusiastically in favor of publishing this work. Also, I have a couple of minor critical points/suggestions, which the authors might wish to address:

Comments, suggestions, and questions to the authors:

1. Lines 325-328: I would like to suggest to the authors to revise this sentence to read something like "However, while bacterial RMF, SHPF, RaiA (protein Y) or the LHPF N-terminal domain all target the mRNA path and the decoding center of the small subunit, eukaryotic Lso2 and CCDC124 bind to both, the 40S and 60S subunits simultaneously and exclusively stabilize the non-rotated conformation of the 80S ribosome."

SHPF - short HPF, LHPF - long HPF. I am not sure what the authors meant by IHPF? Perhaps it was just a typo. Also, RaiA is better known as ribosomal protein Y, so maybe somehow include this name as well?

2. Figure 5, panels C and D: Why the positions of the peaks corresponding to the unsplit 80S ribosomes in panel C do not match the same peaks in panel D (except for the plots with Stm1)? In other words, the 80S peaks at the top and bottom plots should have the same positions relative to the gradient start.

---

## [Editor Report · Decision Letter 2]

1 Jul 2020

Dear Dr Beckmann,

On behalf of my colleagues and the Academic Editor, Jamie H. D. Cate, I am pleased to inform you that we will be delighted to publish your Research Article in PLOS Biology. 

Early Version

PRESS 

Kind regards,

Pamela Berkman

Publishing Editor

PLOS Biology

on behalf of

Roland Roberts,

Senior Editor

PLOS Biology